# Delta 9-tetrahydrocannabinol conserves cardiovascular functions in a rat model of endotoxemia: Involvement of endothelial molecular mechanisms and oxidative-nitrative stress

Bálint Bányai[1], Csaba Répás[2,3,4], Zsuzsanna Miklós[2,5,6], Johnny Johnsen[1], Eszter M. Horváth[1,2]*, Rita Benkő[1,2]

1 Department of Physiology, Semmelweis University, Budapest, Hungary, 2 Institute of Human Physiology and Clinical Experimental Research, Semmelweis University, Budapest, Hungary, 3 Albert Schweitzer Hospital, Hatvan, Hungary, 4 Hungarian National Ambulance Service, Salgótarján, Hungary, 5 Institute of Translational Medicine, Semmelweis University, Budapest, Hungary, 6 National Koranyi Institute for Pulmonology, Budapest, Hungary

* horvath.eszter@med.semmelweis-univ.hu

## Abstract

In endotoxemic models, the inflammatory parameters are altered to a favorable direction as a response to activation of cannabinoid receptors 1 and 2. The phytocannabinoid $\Delta^9$-tetra-hydrocannabinol (THC) is an agonist/partial antagonist of both cannabinoid receptors. This report targets the effects of THC on the cardiovascular system of endotoxemic rats. In our 24-hour endotoxemic rat model (E. coli derived lipopolysaccharide, LPS i.v. 5mg/kg) with THC treatment (LPS+THC 10 mg/kg i.p.), we investigated cardiac function by echocariography and endothelium-dependent relaxation of the thoracic aorta by isometric force measurement compared to vehicle controls. To evaluate the molecular mechanism, we measured endothelial NOS and COX-2 density by immunohistochemistry; and determined the levels of cGMP, the oxidative stress marker 4-hydroxynonenal, the nitrative stress marker 3-nitro-tyrosine, and poly(ADP-ribose) polymers. A decrease in end-systolic and end-diastolic ventricular volumes in the LPS group was observed, which was absent in LPS+THC animals. Endothelium-dependent relaxation was worsened by LPS but not in the LPS+THC group. LPS administration decreased the abundance of cannabinoid receptors. Oxidative-nitrative stress markers showed an increment, and cGMP, eNOS staining showed a decrement in response to LPS. THC only decreased the oxidative-nitrative stress but had no effect on cGMP and eNOS density. COX-2 staining was reduced by THC. We hypothesize that the reduced diastolic filling in the LPS group is a consequence of vascular dysfunction, preventable by THC. The mechanism of action of THC is not based on its local effect on aortic NO homeostasis. The reduced oxidative-nitrative stress and the COX-2 suggest the activation of an anti-inflammatory pathway.

**Data Availability Statement:** All relevant data are within the paper and its Supporting information files.

**Funding:** This work was supported by the Hungarian National Research, Development and Innovation Office NKFIH-FK129206 and by Semmelweis University. The funders had no role in study design, data collection and analysis, decision to publish, or preparation of the manuscript.

**Competing interests:** The authors have declared that no competing interests exist.

## 1. Introduction

In recent years, the effects of endo- and phytocannabinoids have become a field of increasing research, both for their therapeutic and recreational use.

The well-known signaling molecules of the endocannabinoid system are anandamide and 2-arachidonylglycerol. These signaling substances act through cannabinoid receptors 1 and 2 (CB1R and CB2R), which are G protein-coupled receptors, coupled to $G_{i/0}$ protein. CB1R may also be involved in $G_s$-dependent regulation. CB1R is a common receptor found on the surface of neurons and well known by its psychostimulant effects. In addition, it has cardiovascular effects [1–3].

CB2R activation is not a psychostimulant, but CB2R mediated anti-inflammatory effects have been identified that make it a favorable therapeutic target. It is found on the surface of many cells in the periphery, e.g.: on white blood cells, on the surface of skeletal and smooth muscle cells, hepatocytes, and platelets [4].

Recent research has shown that the activity of the endocannabinoid system shows different levels in health and in pathological conditions. Endocannabinoid system shows a characteristic protective activity under pathological conditions. The cardiovascular impact of cannabis has been an emerging area of research in recent years. On the other hand, cardiovascular research shows that stimulation of cannabinoid receptors in healthy individuals does not provide benefits. However, patients with inflammatory diseases, diabetes, obesity, sepsis, protective effects were observed, most of which were due to CB2R activation [4, 5].

The first known natural cannabinoid that was isolated from the hemp (Cannabis sativa) is $\Delta^9$-tetrahydrocannabinol (THC), which is a partial agonist of both CB1R and CB2R [6]. This property allows us to investigate both CB1R and CB2R signaling pathways together. In addition, recent research [6] has shown that THC and other phytocannabinoids act not only through cannabinoid receptors but also through other signaling pathways, including PPARγ receptor (metabolic effects), as agonists of various TRPV channels, particularly via the TRPV1 (capsaicin) receptor (central nervous system effects), and allosteric modulator effects have been observed at glycine and μ/δ-opioid receptors. Agonist effects have been observed also on the newly identified "orphan" receptors; such as GPR-55 ($G_q$ coupled signaling) [7], GPR-18, GPR-119. These receptors like CB receptors belong to the family of G protein coupled receptors [6].

Endotoxemia leads to inflammatory responses and impairs the cardiovascular system, including endothelial dysfunction [8–10] and increase oxidative- nitrative stress. Humans in septic shock have a 50% mortality. Besides an elevated metabolic rate, the adrenergic signaling becomes inefficient, as the liver produces octopamine that masks alpha-adrenergic receptors [11], leading to systemic vasorelaxation. The cardiovascular response is biphasic: first, a hyperdynamic stage occurs with maintained blood pressure and an elevated cardiac output. The second stage starts when the heart cannot maintain the cardiac output necessary for the maintenance of blood pressure in the dilated vasculature. Rodents in sepsis develop cardiac dysfunction characterized by impaired contractility and endothelial dysfunction. Inflammation leads to an elevated (oxygen and nitric-oxide derived) free radical formation. The elevated oxidative and nitrative stress leads to non-specific modifications of lipids and proteins; moreover, single- and double-strain DNA damage also occurs. Mainly the single-strand DNA breaks activate poly(ADP-ribose) polymerase 1 (PARP-1). PARP utilizes $NAD^+$ to build ADP-ribose polymers (PAR), attaches to histones and help repair mechanisms. PAR-ylation refers to a specific protein modification process that alters the cell's self-regulatory mechanisms, such as DNA repair, gene transcription, apoptosis and cell metabolism. However, it can lead to $NAD^+$ depletion, hence, to energetic failure of the cells. The abundance of PAR can be used as

a marker of DNA damage. Reactive oxygen and nitrogen species and the concomitant cellular damage also contribute to the development of cardiovascular diseases [12].

In the present study, we investigated endotoxemia induced cardiovascular damages and the oxidative-nitrative stress parameters in a rat model of endotoxemia combined with THC treatment as a possible therapeutic agent.

## 2. Materials and methods

All investigations conform to the Guide for the Care and Use of Laboratory Animals published by the National Institutes of Health (NIH Publication No. 85–23, Revised 1985) and all procedures were approved by the Semmelweis University Committee on the Ethical Use of Experimental Animals (590/99 Rh).

### 2.1. Animals

Eight age-matched Sprague-Dawley rats weighing 280–350 g were administered a single dose of 10 mg/kg intraperitoneal $\Delta^9$-tertahydrocannabinol (THC) (Sigma Aldrich, St. Louis, MO), solubilized in ethanol:saline: 1:2. 5 to 10 minutes later the animals were administered 5 mg/kg lipopolysaccharide (LPS, from Escherichia coli, Sigma Aldrich, St. Louis, MO) intravenously, suspended in saline. Twelve animals received only the solvent alcohol-saline mix intraperitoneally before the injection of LPS (positive controls), another twelve rats were injected with solvent alcohol-saline mix intraperitoneally and saline intravenously (negative controls). The animals remained in the following 24 hours in their usual environment. As 5 mg/kg LPS causes hyperalgesia, their wellbeing was checked in this period, but they did not receive anesthetics, because anti-inflammatory drugs or opiates would have interfered with the measurements. Echocardiography, cardiovascular measurements and collection of tissue specimens took place 24 hours after LPS treatment under anesthesia (description of anesthesia is provided at the specific measurement). The animals were killed in deep anesthesia by bleeding and opening of the chest wall.

### 2.2. Echocardiography

For echocardiographic studies, animals were superficially anesthetized with 1.3 g/kg urethane. A Hewlett Packard Sonos 5500 ultrasound machine equipped with a 7–15 MHzlinear ultrasound probe was used to capture video loops of 5–6 cardiac cycles. Two-dimensional longitudinal recordings of the left ventricle were used to measure end-systolic volume (ESV), end-diastolic volume (EDV) and stroke volume (SV), as described previously [13, 14]. Ejection fraction was also calculated as SV/EDV. In each experiment, 3 end-systolic and 3 end-diastolic images were selected for analysis, and the average of the 3 measurements was used for further analysis.

The thickness of the left ventricular wall was determined by taking cross-sectional images of the ventricle at the level of the papillary muscles. Fractional shortening (FS) was also determined as the ratio of the shortening in internal diameter during systole and the end-diastolic internal diameter.

The duration of cardiac cycles, and heart rate was determined using M-mode recordings.

### 2.3. Examination of blood pressure and left ventricular function

The animals were anesthetized with i.p. injections of 60 mg/kg thiopentone sodium (Nembutal, Phylaxia-Sanofi, Hungary). The right femoral artery was cannulated for measuring the arterial blood pressure and a catheter were inserted into the left ventricle via the right carotid

artery (PE50 tube, Becton Dickinson, San Jose, CA.) Data were collected and evaluated using Biopac system (Biopac, Goleta, CA, USA). Heart rate (HR), left ventricular systolic and end-diastolic pressure (LVSP and LVEDP) were measured and mean arterial pressure (MAP), left ventricular developed pressure (LVDP) were calculated. In order to estimate ionotropy and lusitropy, the maximal slope of left ventricular contraction (dP/dt) and minimum slope of left ventricular relaxation (-dP/dt) were also calculated.

## 2.4. Measurement of vascular reactivity on isolated aortic rings of rats

The method to determine endothelium-dependent vascular relaxation in thoracic aortic rings from rats was described previously [15]. Briefly, the thoracic aorta was isolated from the barbiturate-anaesthetized rats, cleared from periadventitial fat and cut into 3–4 mm width rings, mounted in organ baths filled with warmed (37 ˚C) and oxygenated (95% O2, 5% CO2 –Carbogen Lindegas) Krebs' solution (CaCl$_2$ 1.6 mM; MgSO$_4$ 1.17 mM; NaCl 130 mM; NaHCO$_3$ 14.9 mM; KCl 4.7 mM; KH$_2$PO$_4$ 1.18 mM; Glucose 11 mM). Isometric tension was measured with isometric transducers (10 cm$^3$ capacity, vertical training organ bath system, Experimetria Ltd. Budapest, Hungary, digitized, stored and displayed by a software developed by Experimetria, Hungary.) A tension of 1.5 gram was applied and the rings were equilibrated for 60 minutes, followed with epinephrine dose-response curve ($10^{-10}$–$3*10^{-6}$ M) and, after a 30 to 60 minute-long washout period, the rings were precontraced with epinephrine ($10^-{}_6$ M) and concentration-dependent relaxation to acetylcholine (Ach, $10^{-9}$ to $3 * 10^-{}_4$ M) was measured. Experiments were conducted in 5–6 pairs of rings in each experimental group.

## 2.5. Malonyl-dialdehyde assay

Serum was isolated for malondialdehyde detection. The samples were stored at -80 ˚C until the time of analysis. Samples were homogenized with 0.5 ml of 1.15% KCl solution and centrifuged at 5000 rpm for approximately 30 min until the supernatant was completely clear. On a standard microplate the following solutions were added to the wells: sodium dodecyl sulfate (8.1%), acetic acid (20%), water, the supernatant, thiobarbituric acid (0.8%) and incubated at 95 ˚C for one hour. Concentration of thiobarbituric acid reactive product was measured by photometry at 532 nm (PowerWave XS, BioTek Instruments, CA, USA.).

## 2.6. Immunohistochemical staining

Immunohistochemistry was performed on paraffin-embedded tissue sections of the thoracic aorta and the heart against poly(ADP-ribose) polymers (PAR), cannabinoid receptor 1 and 2 (CB1R and CB2R); cyclooxygenase-2 (COX-2), endothelial nitric oxide synthase (eNOS) 4-hydroxynonenal (HNE), cyclic guanosine-monophosphate (cGMP) and 3-nitrotyrosine (NT). After deparaffinization, antigens were retrieved by heating the slides in citrate buffer (pH = 3 PAR, CB2R or pH = 6 CB1R, COX-2, eNOS and HNE; for cGMP and NT, we did not apply antigen retrieval). We blocked endogenous peroxidase activity with 3% H$_2$O$_2$ in distilled H$_2$O$_2$. Nonspecific labeling was blocked using 2.5% normal horse serum (Vector Biolabs, Burlingame, CA, U.S.A.). After overnight application of primary antibodies (monoclonal mouse anti-eNOS 1:50, Abcam Cambridge, UK; polyclonal rabbit anti-COX-2 1:200, Abcam Cambridge, UK; polyclonal rabbit anti-cGMP 1: 500, Merck Millipore, Burlington, MA; U.S.A., polyclonal rabbit anti-NT 1: 500, Merck Millipore, Burlington, MA; USA, polyclonal rabbit anti-HNE 1:200, Abcam Cambridge, UK; monoclonal mouse anti-PAR 1:500 Abcam Cambridge, UK; polyclonal rabbit anti-CB1R 1:200, Cayman Chemical, Ann Arbor Michigan U.S.A.; polyclonal rabbit anti-CB2R 1:150, Fabgennix, Thermo Fisher Scientific, Waltham, MA, U.S.A.) at 4˚C, horseradish-peroxidase- (HRP-) linked anti-mouse (PAR, eNOS) or anti-rabbit

(NT, HNE, CB1R, CB2R, cGMP, COX-2) horse antibodies (Vector Biolabs, Burlingame, CA, U.S.A.) provided secondary labeling, which was visualized by brown-colored diamino-benzi-dine (DAB, Vector Biolabs, Burlingame, CA, U.S.A.). For counterstaining, blue-colored hema-toxylin (Vector Biolabs, Burlingame, CA, U.S.A.) was utilized. Nikon Eclipse Ni-U microscope with DS-Ri2 camera (Nikon Minato—Tokyo Japan) was used for microscopic imaging of tis-sue sections. Positively stained area (brown coloring) over whole tissue area (area%) of the endothelium (eNOS and COX-2) and of the media (CB1R, HNE, NT, cGMP); and positively stained nuclear area over whole nuclear area in the media (nuclear area%, PAR) were esti-mated by ImageJ software (NIH, Bethesda, MA, U.S.A.).

### 2.7. Statistical analysis

Results are reported as mean ± Standard Error of the Mean (vascular relaxation) or mean ± Standard Error. Statistical significance between groups was determined by repeated measure two-way analysis of variance (ANOVA) with Bonferroni's multiple comparison (aor-tic rings), one-way ANOVA with Tuckey's multiple comparison (malonyl-dialdehyde assay, cardiac measurements), or Kruskal-Wallis test with Dunn's post hoc test (immunohistochem-istry results). Probability values of $P < 0.05$ were considered significant. All relevant data are within the manuscript and the S1 File.

## 3. Results

### 3.1. Cardiac function

In vivo cardiac functions were examined by echocardiography and left ventricular pressure monitoring. Echocardiography and invasive monitoring showed no significant change in heart rate in the experimental groups. (Fig 1. Panel A, B).

On the other hand, parameters that are heavily influenced by vascular function, such as end-systolic volume (ESV), end-diastolic volume (EDV), and cardiac output (CO) were affected by both LPS and LPS+THC co-treatment (Fig 2). Concerning the other echocardio-graphic parameters, as well as inotropy and lusitropy assessed by invasive methods, no differ-ence could be detected between the studied groups.

### 3.2. Vascular function

Applying $3*10^{-7}$ mol/liter and higher concentrations of acetylcholine (Ach) after epinephrine ($10^{-7}$ mol/liter) precontraction resulted a vascular relaxation distinctly reduced in the LPS

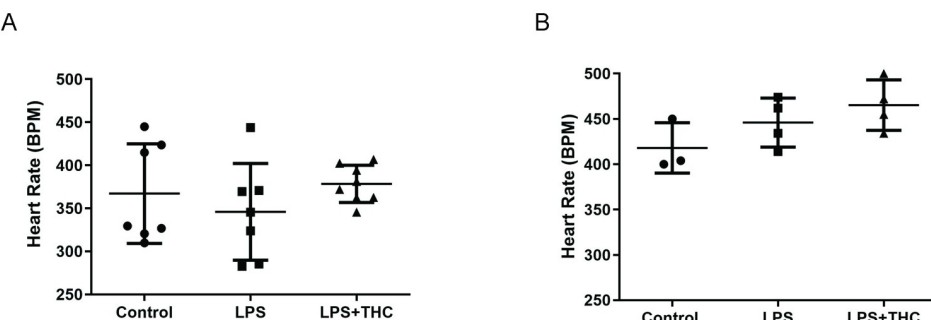

**Fig 1. Heart rate in the three experimental groups (Mean±SD).** Panel A. Heart rate measured in deep thiopentone sodium anesthesia, by an intracardiac catheter (N = 7-8/group). Panel B. Heart rate calculated by cardiac ultrasonography, under urethane anesthesia (N = 3-4/group).

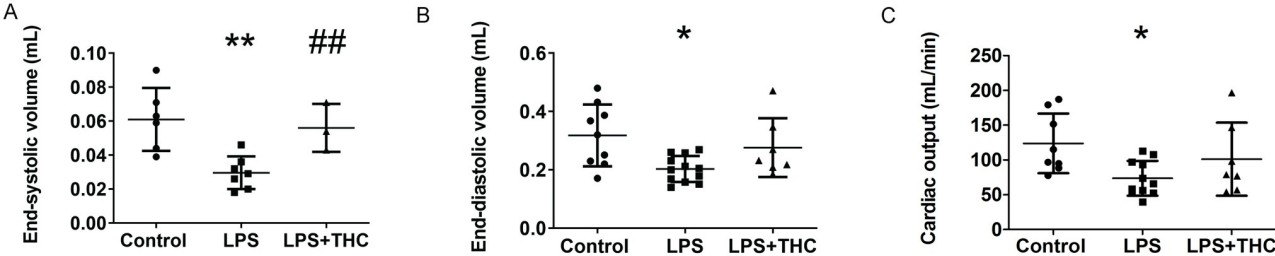

**Fig 2. Heart parameters assessed by cardiac ultrasonography.** Panel A. End systolic volume (ESV). Panel B. End diastolic volume (EDV). Panel C. Cardiac output calculated from EDV and ESV. Individual data points are represented together with mean±SD, one-way ANOVA with Tukey's post-hoc test was performed, *p<0.05 vs. Control, **p<0.01 vs. Control, ##p<0.01 vs. LPS. N = 3-12/group.

group compared to the control vessels. This difference diminished in the LPS+THC group (Fig 3).

## 3.3. Evaluation of oxidative and nitrative stress and consequential DNA-damage

Systemic oxidative stress was assessed by malonyl-dialdehyde assay (MDA), that indicated an elevation of oxidized circulating byproduct in the LPS-treated group (mean±SD: 207.6 ±81.72 μM vs. 67.10±25.21 μM) but the elevation was below the level of significance in the LPS +THC group (111.70±73.32 μM). Tissue oxidative stress was evaluated based on immunohistochemical staining of 4-hydroxy-nonenal in the aortic and cardiac samples. The staining intensity of samples from the LPS group were significantly elevated (optical density in the left ventricle (mean±SD): 0.1965±0.0253 vs. 0.1641±0.0066, p<0.05; positive area in the aortic wall: 4.325±2.725% vs. 0.2943±0.4371, p<0.05), whereas in the LPS+THC rats, this elevation

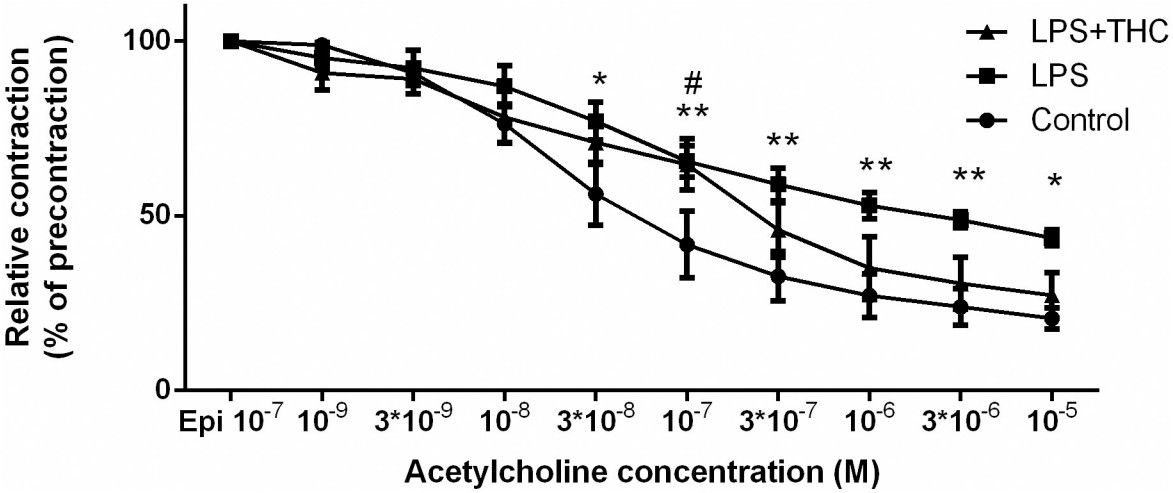

**Fig 3. Acetylcholine-induced relaxation ability of isolated thoracic aorta segments after epinephrine precontraction.** LPS treatment (squares) caused a reduced endothelium-dependent acetylcholine-induced relaxation compared to the Control (circles). THC treatment (triangles) prevented the decrement in the endothelium-mediated relaxation. Data are presented as mean ± SEM, N = 4 in each group; repeated measures ANOVA, Bonnferoni's post hoc test, *p<0.05 Control vs. LPS **p<0.01 Control vs. LPS; #p<0.05 LPS vs. LPS+THC.

was not observable (left ventricular OD (mean±SD): 0.2016±0.0423 and positive area in the aortic wall: 0.0769±0.0683, p<0.05 vs. LPS). (Fig 4. Panel A-C).

In the aortic wall, an elevation of both NT and PAR were detectable only in the LPS group (NT positive area (mean±SD): 20.98±8.064% vs. 11.72±5.337%, p<0.05; PAR positive nuclear area: 0.7215±0.1681% vs. 0.3833±0.1319, p<0.01). Interestingly, THC not just prevented the elevation of PAR-ylation, but it even showed a tendency to decrease nitrative stress in comparison to the controls (NT: 4.744±3.757 p<0.01 vs. LPS; PAR: 0.6430±0.0366, ns.) (Fig 4. Panel D-G).

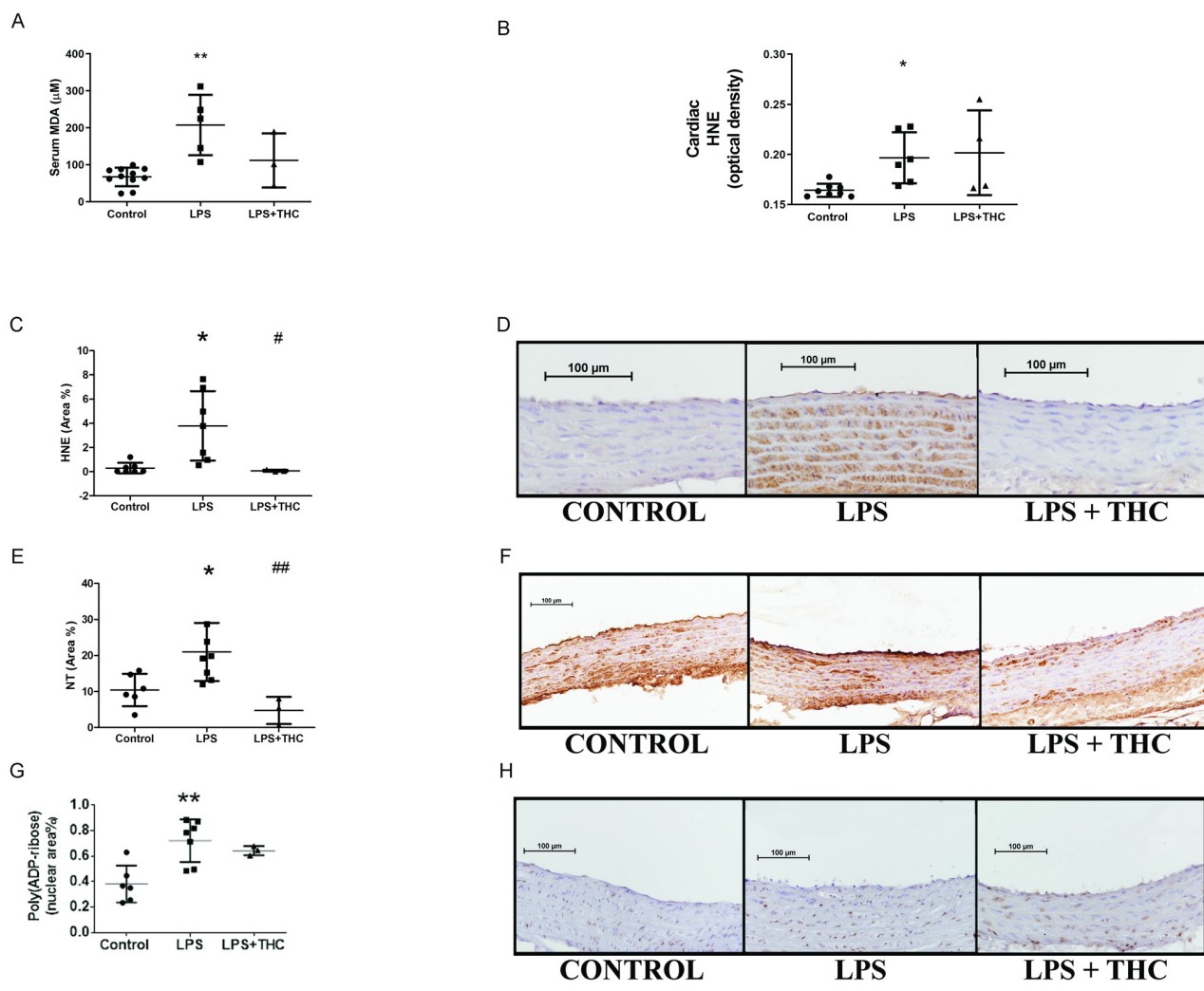

**Fig 4. Systemic and cardiovascular oxidative-nitrative stress.** Panel A.: Systemic oxidative stress detected by malonyl-dialdehyde assay (N = 3–11) was significantly elevated in endotoxemic rats but not in THC-treated animals. **p<0.01 vs. Control. Panel B.: Cardiac (left ventricular) oxidative stress detected by 4-hydroxy-nonenal staining (HNE, optical density, N = 4-7/group) was elevated in endotoxemic rats. The THC-treated animals statistically did not differ from Controls or LPS-treated rats. *p<0.05 vs. Control. Panel C.: 4-hydroxy-nonenal staining (positive area%, N = 4-7/group) in the thoracic aortic wall showed a significant elevation in LPS group, but not in LPS+THC animals. *p<0.05 vs. Control, #p<0.05 vs. LPS. Panel D.: Representative photos of the aortae stained against HNE. Panel E.: Assessing nitrative stress. 3-nitrotyrosine (NT, positive area%, N = 3-8/group) in the thoracic aortic wall showed a significant elevation in LPS group, but not in LPS+THC animals. *p<0.05 vs. Control, ##p<0.01 vs. LPS. Panel F.: Representative photos of the aortae stained against NT. Panel G.: DNA damage assessed by Poly(ADP-ribose) polymer nuclear density (PAR, positive nuclear area%, N = 3-7/group) in the thoracic aortic wall showed a significant elevation in LPS group, but not in LPS+THC animals. **p<0.01 vs. Control. Panel F.: Representative photos of the aortae stained against NT. Panels D., F. and H.: Brown precipitate represents positive staining with violet hematoxilyn counterstaining. Photos were taken with 20-fold magnification. Statistical analysis was executed with Kruskal-Wallis test & Dunn's post-hoc test.

## 3.4. Molecular contributors of vascular endothelial relaxation

The density of endothelial NO synthase (eNOS) was not affected by LPS, but it significantly decreased in the LPS+THC group compared to the control group (Positive area% of Control: 8.826±6.682%, LPS: 3.173±2.373%, LPS+THC*: 2.982±4.116%, *p<0.05 vs. Control, Fig 5. Panel A, B). The inducible form of cyclooxygenase (COX-2) density shows a similar tendency: the LPS+THC groups staining intensity was significantly reduced compared to the Control group (Area%: Control: 16.36±7.134%, LPS: 13.65±8.006%, LPS+THC*: 4.161±1.780%, *p<0.05 vs. Control, Fig 5. Panel C, D). On the other hand, the cyclic guanosine monophosphate (cGMP) staining significantly decreased in the aortic wall of the LPS and the LPS+THC rats compared to the Control group (area%: 1.415±1.556% and 1.934±1.305% vs. 13.49 ±10.35%, p<0.05 Fig 5. Panel E, F). On the contrary, cGMP staining in the left ventricular samples and small coronary vessels were almost identical in all three groups (Optical density of cardiac muscle and coronary arteries MEAN±SD: 0.06101±0.0108 and 0.05793±0.0074 (Control); 0.05735±0.0088 and 0.05630±0.0079 (LPS); 0.05262±0.0033 and 0.05710±0.007 (LPS+THC); P = 0.35 and 0.83).

In case of CB1R, the specific staining indicated a reduction of density in the LPS group (0.06±0.11%) and THC-treated rats (0.2991±0.513%) in comparison to Controls (1.681 ±1.393%, p<0.05 vs. LPS and LPS+THC, Fig 6. Panel A, B). Cannabinoid receptor 2 expression also showed a strong declining tendency in LPS group (0.047±0.05%) in comparison to Controls (0.1876±0.18 area%, p = 0.0548). However, unlike CB1R staining, CB2R abundance was similar in THC-treated animals (0.2215±0.26%) to Controls (Fig 6. Panel C, D).

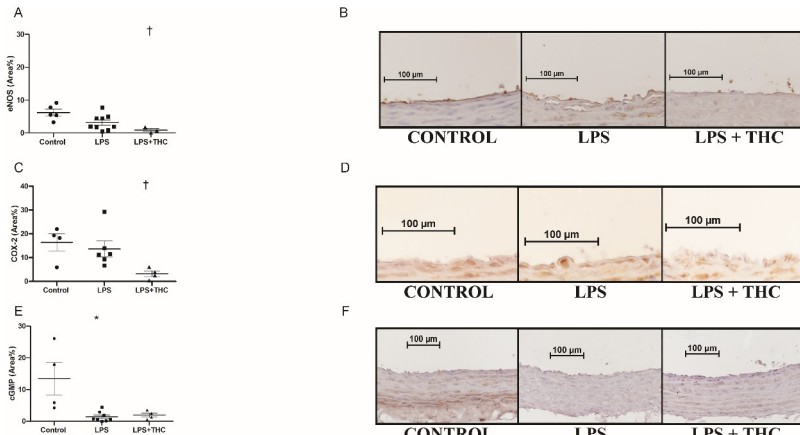

**Fig 5. Histological changes of the vasoactive markers in the thoracic aorta.** (A) endothelial NO-synthase density of the aorta endothelial layer. Data shown by positive area % of the whole aorta segment with mean ±SEM N = 5 (Control)-9 (LPS)-3 (LPS+THC) in the groups †:p<0,05 Control vs. LPS+THC. Statistical analysis were executed with Kruskal-Wallis test & Dunn"s post-hoc test. (B) Representative photos of eNOS stained aorta segments in the endothelial layer, visualization with diamino-bensidine (DAB) on hematoxylin counterstaining, photographed by two-hundredfold magnification. (C) COX-2 density the aorta segments. Data shown by positive area % of the vessels endothelial layer with mean ±SEM; N = 4(Control)-6(LPS)-4(LPS+THC) †:p<0,05 Control vs. LPS+THC. Statistical analysis was executed with Kruskal-Wallis test & Dunn's post-hoc test. (D) Representative photos of COX-2 labelled aorta segments in the endothelial layer, visualization with diamino-bensidine (DAB) on hematoxylin counterstaining, photographed by two-hundredfold magnification. (E) cGMP density of the aorta segments. Data shown by positive area % of the whole aorta segment with mean ±SEM; N = 4(Control)-7(LPS)-4(LPS+THC) *:p<0,05 Control vs. LPS. Statistical analysis were executed with Kruskal-Wallis test & Dunn's post-hoc test. (F) Representative photos of cGMP labelled aorta segments in the endothelial layer, visualization with diamino-bensidine (DAB) on hematoxylin counterstaining, photographed by two-hundredfold magnification.

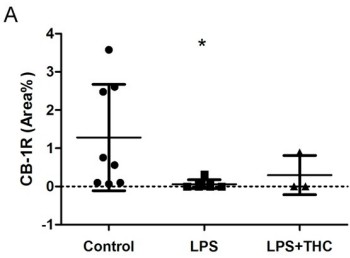

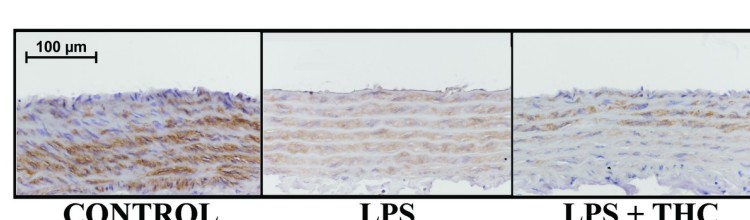

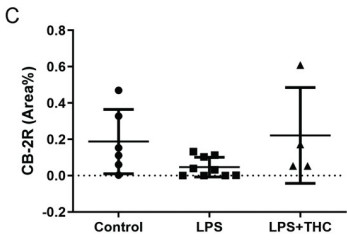

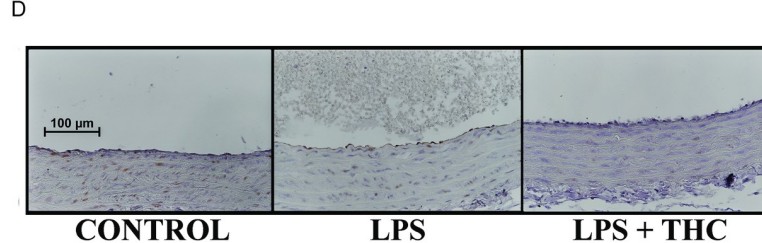

**Fig 6. Histological changes of the cannabinoid receptor 1 and 2 intensity in the thoracic aorta.** (A) CB1R density of the aorta segments. Data shown by positive area % of the whole aorta segment with mean ±SEM n = 8-7-3 in the groups *:p<0,05 Control vs. LPS. (B) Representative photos of CB1R stained aorta segments The positivity labeled with 3' diaminobensidine with brown precipitate, and a violet color hematoxyilin counterstaining, two-hundredfold magnification. (C) CB2R density of the aorta segments. Data shown by positive area % of the whole aorta segment with mean ±SEM n = 6-9-4 in the groups. (D) Representative photos of CB2R stained aorta segments The positivity labeled with 3' diaminobensidine with brown precipitate, and a violet color hematoxyilin counterstaining, two-hundredfold magnification.

## 4. Discussion

In our model, LPS caused a marked decrement in the abundance of CB1R and a strong tendency decreasing CB2R. Nevertheless, Δ⁹-tetrahydrocannabinol salvaged the cardiovascular functions of endotoxemic rats, decreased oxidative-nitrative stress and the detectable amount of inducible cyclooxygenase, but the restored endothelium-mediated relaxation was not dependent on endothelial nitric oxide synthase abundance or cGMP levels.

### 4.1. Cardiac functions in endotoxemia

In the clinics, changes in the circulation occurs in two steps in patients with sepsis. At first, the generalized inflammation leads to systemic vasorelaxation, but the heart is able to maintain the cardiac output necessary for proper tissue perfusion (hyperdynamic stage). The exhaustion of the heart leads to the second phase, the blood pressure decreases and the tissue perfusion becomes insufficient (hypodynamic stage). However, 5 mg/kg LPS is below the lethal dose in rodents [16]. In our experiments, the lethality of LPS was 0% (no animal died within the 24-hour waiting period), and the LPS-inoculated mice did not show signs of severe distress (no anxiety, their eyes were clear and their fur was smooth.)

Our model pointed toward vascular damages behind the compromised ventricular filling and cardiac output; therefore, we propose that the alteration of cardiac parameters (EDV and ESV) are the consequences of large vessel dysfunction. The reduced cardiac output can be the result of decreased ventricular filling. Previous studies performed on murine and rat models revealed a decrement in inotropy in similar models of endotoxemia, utilizing 5 mg/kg LPS. However, in these studies only a four- to six-hour long period was assessed [17–19].

Peng et al. used an LPS dosage of 4 mg/kg and investigated the effect after 4 and 24 hours' incubation [20]. The heart rate did not change due to LPS treatment in either of these studies. Pacher et al. also found ambivalent results—based on their review, THC can be cardio protective in low dosage in a dose dependent manner, or harmful to the cardiac system in higher dosages [21, 22].

In the cardiac samples, only the oxidative stress marker hydroxy-nonenal was elevated, concomitant to systemic elevation of oxidative stress. Therefore, our results are findings can be the consequences of an altered vascular function, and direct cardiac injury cannot be established.

In endotoxemia, an activation of the renin-angiotensin system occurs and the biologically active angiotensin II level elevates; however, the vascular sensitivity to angiotensin decreases [23–25]. The lack of CB1R leads to increased vasoconstriction during acute angiotensin II stimulation [3, 26], hence angiotensin-mediated vasoconstriction may have been altered by THC. The cardiac effects of angiotensin-related cannabinoid signaling are dual, as it influences inotropy and coronary blood flow simultaneously [27]. The activation of CB2R showed beneficial effects in cerebral ischemia-reperfusion [28], neuroinflammation after traumatic brain injury [29], or in rheumatoid arthritis [30, 31]. The CB1R antagonist Rimonabant was cardioprotective in a rodent model of myocardial infarction [32] and decreased matrix metalloproteinase activity after spinal cord injury [33].

## 4.2. Vasoprotective effects of $\Delta^9$-tetrahydrocannabinol

In our study THC treatment was able to restore Ach sensitivity of thoracic aortic rings that had been damaged by LPS treatment, suggesting that THC may ameliorate LPS induced endothelial dysfunction. However, we also detected a decrement in both CB1R and CB2R abundance in the aortic wall as a response to LPS challenge.

Navarro Dorado *et al.* assessed the effects of chronic nonselective CB agonist and antagonist (WIN55 & JWH133) dosage in a transgenic model of Alzheimer's disease. In their model, nonselective CBR activation restored the acetylcholine-induced relaxation [34].

O'Sulivan et al. found that THC had a time dependent vasorelaxing effect (after an acute vasoconstictor effect) through PPAR-γ activation, which is not CB1R dependent [35]. On the other hand, according to Stanley *et al.* canabidiol (CBD), which is another phytocannabinoid, also causes time dependent vasorelaxation but in this case it is CB1R mediated and endothelium-dependent [36]. The different results may be due to the fact that while the former case used a healthy rat model, the latter used human samples from poly-morbid patients. The cannabinoid system is also involved in the regulation of the blood flow of the brain during hypoxia and hypercapnia [37]. However, in these models, the research assessed the direct vasorelaxant effects of cannabinoid agonists, whereas, in our study, the indirect vasoprotective effect of THC was detected, as THC has a short half-life of only 2 minutes, when administered intravenously; and because the downregulation of the cannabinoid receptors.

Other studies about streptozotocin induced diabetic cardiomyopathy showed similar results after chronic cannabinoid treatment. The Ach induced vasorelaxation also decreased in the streptozotocin-induced group and THC was able to restore this vasodilator capability. According to their results, the trends in oxidative-nitrative stress was comparable to our findings: streptozotocin significantly increased lipid peroxidation and nitrative stress markers, which were restored by THC [38, 39].

The low eNOS and cGMP levels and the maintained endothelium-dependent relaxation in the LPS+THC group are contradicting findings. The relaxation of a vessel depends on the balance between vasoconstrictor and vasodilator messengers. The decreased COX-2 detectability may indicate a reduced thromboxane A2 (TxA2) production in the THC-treated group;

therefore, even with a decreased NO bioavailability, the aortic relaxation may be maintained in vitro.

In vivo, the augmented ventricular filling may be the result of the maintained vascular function due to the controlled oxidative-nitrative stress and the absence of elevated TxA2 release from the endothelial cells and platelets, as thrombocyte function is also altered in endotoxemia. In the presence of adenosine diphosphate, LPS-challenged platelets release hydrogen-peroxide and TxA2 [40, 41]. Upon activation, platelets and macrophages may also contribute to the developing hypotension in septic state by releasing 2-arachidonyl glycerol and anandamide; the hypotension was proven preventable with CB1R antagonists [42]. Furthermore, chronic marijuana abuse leads to an increased risk of thrombus formation by platelet activation; however, the basis of the thrombosis is strongly connected to cannabis arteritis [43].

## 4.3. Anti-inflammatory effects of $\Delta^9$-tetrahydrocannabinol

In our model, the aortic density of the inducible cyclooxygenase (COX-2) and nitrative stress (detected by 3-nitrotyrosine staining) were decreased in the THC-treated rats. As the endothelial nitric oxide synthase was also decreased along with the cGMP levels, the results suggest a decreased activity of inducible nitric oxide synthase (iNOS). These results are supported by numerous studies, first mentioning the study by Joffre *et al.* In a complex experiment involving several knock-out strains, the effects of THC and other cannabinoid agonists were monitored during LPS treatments at different doses, and the results showed that cannabinoids significantly decreased the concentration of proinflammatory cytokines (IL-6, CCL-2) and increased the concentration of anti-inflammatory cytokines (IL-10) [44]. These anti-inflammatory properties of THC were also suggested by Suryavanshi *et al.*, on lipopolysaccharide-induced inflammatory response in human THP-1 macrophages and primary human bronchial epithelial cells [45] and Szekely *et al.* on LPS-challenged whole human blood cells [46]. In the latter two models, a cytokine storm-like response developed that was successfully ameliorated by THC. Similar, promising results came forward in the latter years, suggesting the potential immunomodulatory properties of cannabinoids, which may have therapeutic potential [47–49].

## 4.4. Changes in cannabinoid receptor 1 density in endotoxemia

CB1R is involved, as it is highly expressed within the cardiac and vascular cells, as well as on the endothelial cells and a suggested vascular smooth muscle receptor for cannabinoids. Our results suggest a decreased cannabinoid receptor 1. and 2. presence in the aorta during endotoxemia. The downregulation of biologically available cannabinoid receptors can be a result of decreased expression or increased receptor turnover.

Another important mode of signaling is through alternation by β-arrestin2, leading to receptor internalization and desensitization of the cells. Hunyady and his team discovered differences in affinity to β-arrestin2 of cannabinoid receptor 1 and 2 [50, 51], in two different missense polymorphisms of CB2R. One of the mutant receptors showed decreased affinity to β-arrestin2. In case of the control rats, the relatively large deviation in cannabinoid receptor density may be a result of a genetic polymorphism in the receptor transcription, trafficking or turnover.

Differences between mutants and wild-type CB2R within the population causes changes in cAMP levels, as well as downstream MAPK/PI3K signaling, especially when stimulated for an extended amount of time, probably due to receptor internalization and trafficking. Further on, this may play an important role in the endocannabinoid system response, as well as in the pathogenesis of various diseases.

In endotoxemia, the involvement of the cannabinoid system is presented. An elevation of macrophage- and platelet-derived endogenous cannabinoid concentration can be detected, that may contribute to the hemodynamic changes via CB1R activation [42, 52–54]; on the other hand, it may also reduce leukocyte adhesion by a CB2R mediated pathway [55]. Grunewald et al. even suggested that an increment of endogenous cannabinoid production contributes to the LPS-mediated insulin resistance in obesity [56]. However, downregulation of CB1R density in the large arteries of endotoxemic animals was not detected, because it was not investigated, as studies usually target the amount of the endogenous ligands and not the receptor density. Although, in T-cells, CB1R transcription elevated as a response to THC treatment [57]. The decreased CB1R density may explain, why our rats needed 10 mg/kg THC, instead of 2 mg/kg, and similar studies also use comparable doses of THC [44, 58].

## 5. Conclusion

The presented results support the notion that a non-selective CB1/2R agonist–partial antagonist may have therapeutic potential in the treatment of sepsis. In our model, the decrement of cardiac filling and the consequential decline of the cardiac output was prevented by THC treatment, due to the maintained endothelial function. One possible mechanism of the more pronounced endothelium-mediated vasodilation is the decreased thromboxane A2 release due to the lessened inducible cyclooxygenase expression, the other salvaging mechanism is the dampened oxidative-nitrative stress. The activation of endocannabinoid system in inflammation and endotoxemia was earlier described; however, the diminished abundance of both cannabinoid receptors in endotoxemia was not detected. The decreased oxidative-nitrative stress and DNA damage are potentially beneficial in a systemic inflammation, and the reduced inflammatory response may help in the prevention to a quick and robust pro-inflammatory cytokine release (cytokine storm).

## Supporting information

**S1 File.** Contains all collected data that were the basis of all statistical analyses, organized according to the figures, in the following order: Fig 1. Heart rate in the three experimental groups (Panel A: heart rate from invasive measurement (pressure); Panel B: heart rate from cardiac ultrasonography). Fig 2. Heart parameters assessed by cardiac ultrasonography (Panel A: end-systolic volume; Panel B: end-diastolic volume; Panel C: end-diastolic volume). Fig 3. Acetylcholine-induced relaxation ability of isolated thoracic aorta segments after epinephrine precontraction. Fig 4. Systemic and cardiovascular oxidative-nitrative stress (Panel A. MDA; Panel B.Cardiac (left ventricular) oxidative stress detected by 4-hydroxy-noneal staining; Panel C. 4-hydroxy-noneal staining of the aorta; Panel E. Assessing nitrative stress. 3-nitrotyrosine staining in the aorta; Panel G. PAR staining of the aorta). Fig 5. Histological changes of the vasoactive markers in the thoracic aorta (Panel A. eNOS staining of the endothelium; Panel C. COX-2 staining of the endothelium; Panel E. cGMP in the aortic wall). Fig 6. Histological changes of the Cannabinoid receptor 1. and 2. intensity in the thoracic aorta (CB1R; CB2R). (XLSX)

## Author Contributions

**Conceptualization:** Csaba Répás, Zsuzsanna Miklós, Eszter M. Horváth, Rita Benkő.

**Data curation:** Bálint Bányai, Csaba Répás, Zsuzsanna Miklós, Johnny Johnsen, Eszter M. Horváth, Rita Benkő.

**Formal analysis:** Johnny Johnsen, Rita Benkő.

**Investigation:** Bálint Bányai, Csaba Répás, Zsuzsanna Miklós, Johnny Johnsen, Rita Benkő.

**Methodology:** Csaba Répás, Zsuzsanna Miklós, Rita Benkő.

**Supervision:** Eszter M. Horváth, Rita Benkő.

**Validation:** Rita Benkő.

**Visualization:** Bálint Bányai, Csaba Répás, Rita Benkő.

**Writing – original draft:** Bálint Bányai, Zsuzsanna Miklós, Johnny Johnsen, Rita Benkő.

**Writing – review & editing:** Csaba Répás, Zsuzsanna Miklós, Eszter M. Horváth, Rita Benkő.

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
