## [Decision Letter · Decision Letter 0]

14 Feb 2023

PONE-D-22-32081Delta 9-tetrahydrocannabinol conserves cardiovascular functions in a rat model of endotoxemia: involvement of endothelial molecular mechanisms and oxidative-nitrative stress.PLOS ONE

Dear Dr. Horvath,

Thank you for submitting your manuscript to PLOS ONE. After careful consideration, we feel that it has merit but does not fully meet PLOS ONE’s publication criteria as it currently stands. Therefore, we invite you to submit a revised version of the manuscript that addresses the points raised during the review process. Please take into account comments from both reviewers , especially regarding flow and coherency of writing of the manuscript.

We look forward to receiving your revised manuscript.

Kind regards,

Daniel M. Johnson, PhD

Academic Editor

PLOS ONE

Journal Requirements:

2. To comply with PLOS ONE submissions requirements, in your Methods section, please provide additional information regarding the experiments involving animals and ensure you have included details on (1) methods of sacrifice, and (2) efforts to alleviate suffering.

"The author(s) received no specific funding for this work.The study was supported by Semmelweis University."

Reviewers' comments:

Reviewer's Responses to Questions

**Comments to the Author**

1. Is the manuscript technically sound, and do the data support the conclusions?

Reviewer #1: Partly

Reviewer #2: Yes

2. Has the statistical analysis been performed appropriately and rigorously? 

Reviewer #1: Yes

Reviewer #2: Yes

3. Have the authors made all data underlying the findings in their manuscript fully available?

Reviewer #1: Yes

Reviewer #2: Yes

4. Is the manuscript presented in an intelligible fashion and written in standard English?

Reviewer #1: Yes

Reviewer #2: Yes

5. Review Comments to the Author

Reviewer #1: The authors examined the effects of THC on inflammation and cardiac function in a rat model of endotoxemia. They found that vascular function and some measures of cardiac function that were impaired by endotoxin administration were restored by THC administration. They also found that measures of oxidative/nitrative stress were decreased by THC administration in endotoxemia. Finally, they found that CB1R density was reduced in endotoxemia and was not restored by THC administration. The authors clearly put a lot of work into creating this paper, as demonstrated by the wide range of experimental techniques used.

Major comments

The major flaw of this work is the missing control group with THC administration. Also, the group sizes are quite variable, even within the same experiment. Some of the groups are very small, with only 3 or 4 animals represented in the figure. This occurs most frequently in the LPS + THC group, which started out with less animals than the controls (8 versus 12 in each control group). Adding more animals so that the group sizes are consistent would be beneficial.

Minor comments

The link from the experimental model to sepsis is not clearly explained. It is briefly mentioned in the introduction, but few details are given about its effects and importance. If the focus of the paper is treating the cardiac effects of sepsis, this should be evident from the beginning of the paper and the cardiac changes that occur during sepsis should be highlighted in the introduction.

THC activation of CB1R and CB2R was discussed throughout the paper, but only CB1R expression was ever examined. Explaining why CB2R expression was not examined would be helpful in understanding the experimental model.

The writing, especially the introduction and discussion, should be revised to flow more logically and clearly connect the different ideas discussed. Some of the transitions between paragraphs are abrupt and the connection between concepts is not clear until part of the way through the text.

Reviewer #2: The manuscript adds significantly to the research field. The importance of THC to conserve cardiovascular functions in endotoxemia model was well shown.

Some minor changes should be done or information added:

1. Line 368: please, correct the word "vasorelaxation".

2. The importance of platelets in sepsis condition is well stablished ("Lipopolysaccharide treatment reduces rat platelet aggregation independent of intracellular reactive-oxygen species generation. doi: 10.3109/09537104.2011.603065). A paragraph about platelet number and function and its correlation with THC should be added.

3. It is not clear through which mechanism THC reestablishes the relaxation of vessels if not through nitric oxide release. This should be better explained.

4.It is not clear exactly what THC is, in the abstract. More information about this drug should be in the abstract.

6. PLOS authors have the option to publish the peer review history of their article (what does this mean?). If published, this will include your full peer review and any attached files.

Reviewer #1: No

Reviewer #2: No

---

## [Author Response · Author response to Decision Letter 0]

31 Mar 2023

Journal requirements

We reformatted the manuscript, using the correct headings and subheadings, references, and added financial support. In methods, we included two statements regarding the suffering of the rats:

“The animals remained in the following 24 hours in their usual environment. As 5 mg/kg LPS causes hyperalgesia, their wellbeing was checked in this period, but they did not receive anesthetics, because anti-inflammatory drugs or opiates would have interfered with the measurements.”

“The animals were killed in deep anesthesia by bleeding and opening of the chest wall.”

Response to Reviewers

Reviewer #1: 

The authors examined the effects of THC on inflammation and cardiac function in a rat model of endotoxemia. They found that vascular function and some measures of cardiac function that were impaired by endotoxin administration were restored by THC administration. They also found that measures of oxidative/nitrative stress were decreased by THC administration in endotoxemia. Finally, they found that CB1R density was reduced in endotoxemia and was not restored by THC administration. The authors clearly put a lot of work into creating this paper, as demonstrated by the wide range of experimental techniques used.

Thank you for your thorough review and constructive criticism of the manuscript. 

Major comments

The major flaw of this work is the missing control group with THC administration. Also, the group sizes are quite variable, even within the same experiment. Some of the groups are very small, with only 3 or 4 animals represented in the figure. This occurs most frequently in the LPS + THC group, which started out with less animals than the controls (8 versus 12 in each control group). Adding more animals so that the group sizes are consistent would be beneficial.

Answers to the major comments:

During the preliminary experiments, we introduced 2 mg/kg and 10 mg/kg THC to control and LPS-challenged rats. The measured cardiovascular parameters were not altered 24 hours after THC administration in controls; however, 2 mg/kg THC failed to improve the status of LPS-challenged rats. The data collected from rats that only received 10 mg/kg THC (N=2): End-diastolic volume 0.288 mL, end-systolic volume: 0.078 mL, cardiac output: 133 mL; vasorelaxation did not differ from Controls; therefore, we concluded that the results are not the direct effects of THC, but the dampened harm of LPS. 

We understand that „would be beneficial” here is a polite imperative; however, it could not be executed. The permission for this animal experiment and for the possession of THC both expired since the last in vivo measurements. Furthermore, the wire myograph that was used for the measurements are no longer available, we could only use a myograph produced by a different manufacturer. Therefore, in order to increase the group sizes, we would have needed to start over the entire experiment. 

Minor comments

The link from the experimental model to sepsis is not clearly explained. It is briefly mentioned in the introduction, but few details are given about its effects and importance. If the focus of the paper is treating the cardiac effects of sepsis, this should be evident from the beginning of the paper and the cardiac changes that occur during sepsis should be highlighted in the introduction.

The following paragraph was added to the introduction:

“Humans in septic shock have a 50% mortality. Besides an elevated metabolic rate, the adrenergic signaling becomes inefficient, as the liver produces octopamine that masks alpha-adrenergic receptors(11), leading to systemic vasorelaxation. The cardiovascular response is biphasic: first, a hyperdynamic stage occurs with maintained blood pressure and an elevated cardiac output. The second stage starts when the heart cannot maintain the cardiac output necessary for the maintenance of blood pressure in the dilated vasculature. Rodents in sepsis develop cardiac dysfunction characterized by impaired contractility and endothelial dysfunction.”

THC activation of CB1R and CB2R was discussed throughout the paper, but only CB1R expression was ever examined. Explaining why CB2R expression was not examined would be helpful in understanding the experimental model.

The CB2R staining was also implemented at your suggestion, and we have found a similar dynamic in LPS-challenged aortas. The Control (N=6) showed a 0.1876�0.18 area% staining and LPS (N=9) 0.047�0.05% (MEAN�SD). However, in the LPS+THC group (N=4), staining was 0.2215�0.26%. According to the Kruskal-Wallis test, the probability of the three group being samples from the same population is 5.48%, showing a strong tendency. Accordingly, we included this result in the manuscript, and drew the conclusion that LPS decreases the expression of both CB1 and CB2 receptors in the aorta. 

Accordingly, we added to the Results, Figure Legends, Discussion and Conclusion:

“Cannabinoid receptor 2 expression also showed a strong declining tendency in LPS group (0.047�0.05%) in comparison to Controls (0.1876�0.18 area%, p=0.0548). However, unlike CB1R staining, CB2R abundance was similar in THC-treated animals (0.2215�0.26%) to Controls (Figure 6. Panel C-D.).” (MS)

“(C) CB2R density of the aorta segments. Data shown by positive area % of the whole aorta segment with mean ±SEM n=6-9-4 in the groups. (D) Representative photos of CB2R stained aorta segments The positivity labeled with 3’ diaminobensidine with brown precipitate, and a violet color hematoxyilin counterstaining, two-hundredfold magnification.” (FL)

“However, we also detected a decrement in both CB1R and CB2R abundance in the aortic wall as a response to LPS challenge.” and “However, in these models, the research assessed the direct vasorelaxant effects of cannabinoid agonists, whereas, in our study, the indirect vasoprotective effect of THC was detected, as THC has a short half-life of only 2 minutes, when administered intravenously; and because the downregulation of the cannabinoid receptors.” (D 4.2)

“The activation of endocannabinoid system in inflammation and endotoxemia was earlier described; however, the diminished abundance of both cannabinoid receptors in endotoxemia was not detected.” (C) 

The writing, especially the introduction and discussion, should be revised to flow more logically and clearly connect the different ideas discussed. Some of the transitions between paragraphs are abrupt and the connection between concepts is not clear until part of the way through the text.

We edited the Introduction: starting with the cannabinoids, and then describing endotoxemic response in general. We implemented changes through the Discussion to make it more logical and comprehensive. 

Reviewer #2: The manuscript adds significantly to the research field. The importance of THC to conserve cardiovascular functions in endotoxemia model was well shown.

Some minor changes should be done or information added:

Thank you for reading the presentation and for your comments.

1. Line 368: please, correct the word "vasorelaxation".

It is corrected, thank you for noticing the typo. 

2. The importance of platelets in sepsis condition is well established ("Lipopolysaccharide treatment reduces rat platelet aggregation independent of intracellular reactive-oxygen species generation. doi: 10.3109/09537104.2011.603065). A paragraph about platelet number and function and its correlation with THC should be added. 3. It is not clear through which mechanism THC reestablishes the relaxation of vessels if not through nitric oxide release. This should be better explained.

Thank you for the remark, our team is also surprised by this result. The presented measurements do not give a complete answer to the question, we suggest that the reduced oxidative-nitrative stress helps with a more efficient NO handling; also, the decrement in COX-2 abundance leads to the presumption that some vasoconstrictor agents, especially thromboxan A2 production is reduced. The manuscript was amended accordingly:

„The low eNOS and cGMP levels and the maintained endothelium-dependent relaxation in the LPS+THC group are contradicting findings. The relaxation of a vessel depends on the balance between vasoconstrictor and vasodilator messengers. The decreased COX-2 detectability may indicate a reduced thromboxane A2 (TxA2) production in the THC-treated group; therefore, even with a decreased NO bioavailability, the aortic relaxation may be maintained in vitro.

In vivo, the augmented ventricular filling may be the result of the maintained vascular function due to the controlled oxidative-nitrative stress and the absence of elevated TxA2 release from the endothelial cells and platelets, as thrombocyte function is also altered in endotoxemia. In the presence of adenosine diphosphate, LPS-challenged platelets release hydrogen-peroxide and TxA2(39, 40). Upon activation, platelets and macrophages may also contribute to the developing hypotension in septic state by releasing 2-arachidonyl glycerol and anandamide; the hypotension was proven preventable with CB1R antagonists(41). Furthermore, chronic marijuana abuse leads to an increased risk of thrombus formation by platelet activation; however, the basis of the thrombosis is strongly connected to cannabis arteritis(42).” (Discussion 4.2)

“One possible mechanism of the more pronounced endothelium-mediated vasodilation is the decreased thromboxane A2 release due to the lessened inducible cyclooxygenase expression, the other salvaging mechanism is the dampened oxidative-nitrative stress.” (Conclusion)

4.It is not clear exactly what THC is, in the abstract. More information about this drug should be in the abstract.

Thank you for the suggestion, the abstract was complemented with „The phytocannabinoid Δ9-tetrahydrocannabinol (THC) is an agonist partial antagonist of both cannabinoid receptors.”

---

## [Decision Letter · Decision Letter 1]

31 May 2023

Delta 9-tetrahydrocannabinol conserves cardiovascular functions in a rat model of endotoxemia: involvement of endothelial molecular mechanisms and oxidative-nitrative stress.

PONE-D-22-32081R1

Dear Dr. Horvath,

We’re pleased to inform you that your manuscript has been judged scientifically suitable for publication and will be formally accepted for publication once it meets all outstanding technical requirements.

Kind regards,

Daniel M. Johnson, PhD

Academic Editor

PLOS ONE

Additional Editor Comments (optional):

Reviewers' comments:

Reviewer's Responses to Questions

**Comments to the Author**

1. If the authors have adequately addressed your comments raised in a previous round of review and you feel that this manuscript is now acceptable for publication, you may indicate that here to bypass the “Comments to the Author” section, enter your conflict of interest statement in the “Confidential to Editor” section, and submit your "Accept" recommendation.

Reviewer #1: All comments have been addressed

2. Is the manuscript technically sound, and do the data support the conclusions?

Reviewer #1: (No Response)

3. Has the statistical analysis been performed appropriately and rigorously? 

Reviewer #1: (No Response)

4. Have the authors made all data underlying the findings in their manuscript fully available?

Reviewer #1: (No Response)

5. Is the manuscript presented in an intelligible fashion and written in standard English?

Reviewer #1: (No Response)

6. Review Comments to the Author

Reviewer #1: (No Response)

7. PLOS authors have the option to publish the peer review history of their article (what does this mean?). If published, this will include your full peer review and any attached files.

Reviewer #1: No

---

## [Editor Report · Acceptance letter]

8 Jun 2023

PONE-D-22-32081R1 

Delta 9-tetrahydrocannabinol conserves cardiovascular functions in a rat model of endotoxemia: involvement of endothelial molecular mechanisms and oxidative-nitrative stress. 

Dear Dr. Horváth:

I'm pleased to inform you that your manuscript has been deemed suitable for publication in PLOS ONE. Congratulations! Your manuscript is now with our production department. 

Kind regards, 

on behalf of

Dr. Daniel M. Johnson 

Academic Editor

PLOS ONE